# Monomeric prefusion structure of an extremophile gamete fusogen and stepwise formation of the postfusion trimeric state

Juan Feng[1,2], Xianchi Dong[1,2,3], Yang Su[1,2,4], Chafen Lu[1,2] & Timothy A. Springer [1,2 ✉]

Here, we study the gamete fusogen HAP2 from *Cyanidioschyzon merolae* (Cyani), an extremophile red algae that grows at acidic pH at 45 °C. HAP2 has a trimeric postfusion structure with similarity to viral class II fusion proteins, but its prefusion structure has been elusive. The crystal structure of a monomeric prefusion state of Cyani HAP2 shows it is highly extended with three domains in the order D2, D1, and D3. Three hydrophobic fusion loops at the tip of D2 are each required for postfusion state formation. We followed by negative stain electron microscopy steps in the process of detergent micelle-stimulated postfusion state formation. In an intermediate state, two or three linear HAP2 monomers associate at the end of D2 bearing its fusion loops. Subsequently, D2 and D1 line the core of a trimer and D3 folds back over the exterior of D1 and D2. D3 is not required for formation of intermediate or postfusion-like states.

[1] Program in Cellular and Molecular Medicine, Department of Pediatrics, Boston Children's Hospital, Boston, MA, USA. [2] Department of Biological Chemistry and Molecular Pharmacology and Harvard Medical School, Boston, MA, USA. [3]Present address: School of Life Sciences, Nanjing University, Nanjing, China. [4]Present address: Department of Cell Biology, Harvard Medical School, Boston, MA, USA. ✉email: springer@crystal.harvard.edu

Sexual reproduction requires fusion between the plasma membranes of two gametes. The protein HAP2/GCS1 (here called HAP2) is a gamete fusogen that is found in a wide range of eukaryotes including protists, algae, plants, insects, and hemichordates, and may be the "Eve" that enabled sex to evolve in early eukaryotes[1]. Remarkably, HAP2 is structurally related to fusogens present in the membrane of enveloped viruses that allow them to fuse with and infect host cells. Structures of such class II viral fusion proteins are known in two states, a prefusion state, and a trimeric, postfusion state[2,3]. In contrast, all structures to date of the eukaryotic gamete fusogen HAP2, whether from the unicellular green algae *Chlamydomonas reinhardtii* (Chlamy) or the model flowering plant *Arabidopsis thaliana*, are of the trimeric, postfusion conformation[4–7].

Although their sequences are highly divergent, HAP2 and viral class II fusogens have a common overall fold and domain arrangement. Viral class II prefusion structures have three domains, with domain II (D2) most membrane distal, followed by domain I (D1), domain III (D3), a stem, and transmembrane (TM) domain (Fig. 1a). In this arrangement, the hydrophobic fusion loops in D2 are at the end opposite from the C-terminal transmembrane domain. Class II viral fusion proteins themselves are highly divergent, with no detectable sequence homology between fusion proteins of different virus families. In general, prefusion state ectodomains do not extend normally to the viral surface, but lie parallel to the viral membrane envelope. Depending on the virus family, prefusion monomers pack either with one another, such as in anti-parallel dimers, or against other viral surface proteins. During fusion (Fig. 1), prefusion state monomers are thought to come together to form a trimer that

extends away from the viral surface with the hydrophobic fusion loops in D2 at the trimer apex poised for insertion into the host membrane[2,3] (Fig. 1d). However, whether this first stage of trimerization occurs prior to, or after insertion of a monomer into the host membrane, is unknown. The monomers arrange around the trimer threefold axis with D2 and D1 close to the axis. Next, a large bending motion of D3 relative to D1 positions D3 outermost on the trimer axis packed against D1 and D2 (Fig. 1e). The stem between D3 and the TM domain extends toward the D2 fusion loops. This rearrangement "pulls" the viral TM domain and the virus membrane envelope into close proximity with the host membrane for fusion (Fig. 1f).

Little is known about early intermediates in the fusion process (Fig. 1a–e). However, deletion of D3 from a viral class II fusogen showed that the purified construct containing D1 and D2 retained its ability to trimerize, although the postfusion state was less stable[8]. This finding suggested that trimerization of D1–D2 preceded D3 hairpin foldback (Fig. 1d), but earlier intermediates have not been defined for viral fusogens and fusion intermediates in HAP2 are completely undefined.

Here we have determined the high-resolution structure of a prefusion state of HAP2 and low-resolution structures of intermediates in the pathway to its trimeric postfusion state. We study HAP2 from *Cyanidioschyzon merolae* (Cyani), an extremophile, unicellular red alga adapted to high sulfur acidic hot spring environments that grows optimally at pH 1.5 and 45 °C[9]. Our results define the prefusion state of HAP2, which is monomeric with an overall rod-like shape. Cyani HAP2 has unexpected features including an interim β-sheet in D1 that has not been seen in HAP2 postfusion structures or in any state of other class II fusion proteins. Hydrophobic environments in detergent micelles stimulate the formation of a trimeric, postfusion conformational state of Cyani HAP2. The postfusion state is formed at physiological acidic, but not non-physiologic neutral pH. At intermediate values of pH 5.0, we were able to trap Cyani HAP2 in a trimeric state in which three rod-like molecules associate at the D2 end of the rod.

## Results

**The prefusion state of Cyani HAP2.** The Cyani HAP2 ectodomain (aa 28–564) was expressed in *Drosophila* S2 cells. A crystal structure of Cyani HAP2 to 2.3 Å resolution was solved by molecular replacement with the Chlamy postfusion HAP2 structure, and was refined to Rwork and Rfree values of 22.3 and 26.2%, respectively (Supplementary Table 1). Cyani HAP2 crystallizes as a monomer and has a rod-like shape with D3 in line with D2 and D1 (Fig. 2a). It is much more extended than Chlamy HAP2, in which D3 folds back over D2 and D1 in the trimeric, postfusion state (Fig. 2b). These differences are emphasized by the superimposition of one monomer of the Chlamy postfusion trimer on D1 of the prefusion state of Cyani HAP2 (Fig. 2c).

Multi-angle light scattering (MALS) showed that Cyani HAP2 was monomeric in solution (Fig. 2d). PISA assembly analysis of the Cyani HAP2 crystal structure showed that no significant interactions with other monomers were present in the crystal lattice and thus that Cyani HAP2 was monomeric at high concentrations in crystals as well as in solution. These results show that Cyani HAP2 crystallized as a monomeric state distinct from the previous trimeric, postfusion state HAP2 structures[5–7].

Multiple lines of evidence suggest that the monomeric state of Cyani characterized here is a prefusion state. It is well accepted that prefusion states of class II fusion proteins are metastable. Although the stimuli for triggering fusion differ, the postfusion

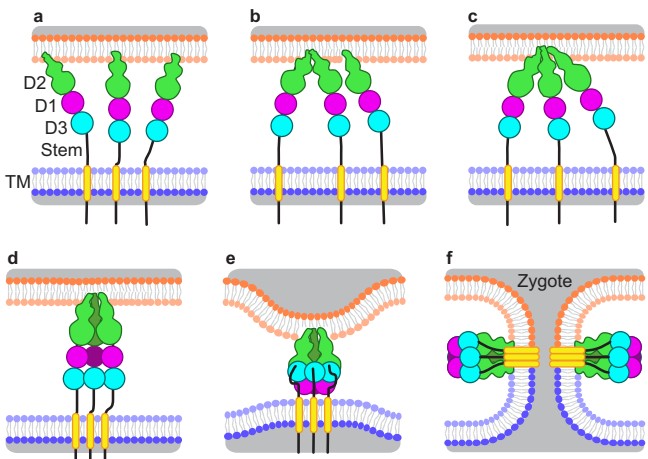

**Fig. 1 Proposed model of the conformational change events in HAP2-mediated membrane fusion. a** Prefusion HAP2 monomers anchored in a gamete membrane (blue bilayer) through their TM domains (yellow rectangles) interacting through their fusion loops in D2 with the gamete of opposite sex (upper orange bilayer). **b** HAP2 dimerization through fusion loop association in a hydrophobic environment. **c** HAP2 dimer and monomer association through fusion loops in a hydrophobic environment initiates trimer formation. **d** Zippering between D1 and D2 of the monomers leads to an extended trimeric conformation. **e** Foldback of D3 pulls the two membranes closer together. **f** In fusion, the stem region (black lines) and TM region associate with the fusion loops during merger of the gamete membranes to open a fusion pore between the two gametes. Intermediates seen in this paper provide evidence for Cyani HAP2 that steps shown in **a–c** precede steps in **d**, **e**. Deletion of D3 shown here and previously[8] provide evidence that zippering of D1 and D2 in **d** can precede D3 foldback in **e**. The steps shown in **e**, **f** are the consensus model in the fusogen field[2–4].

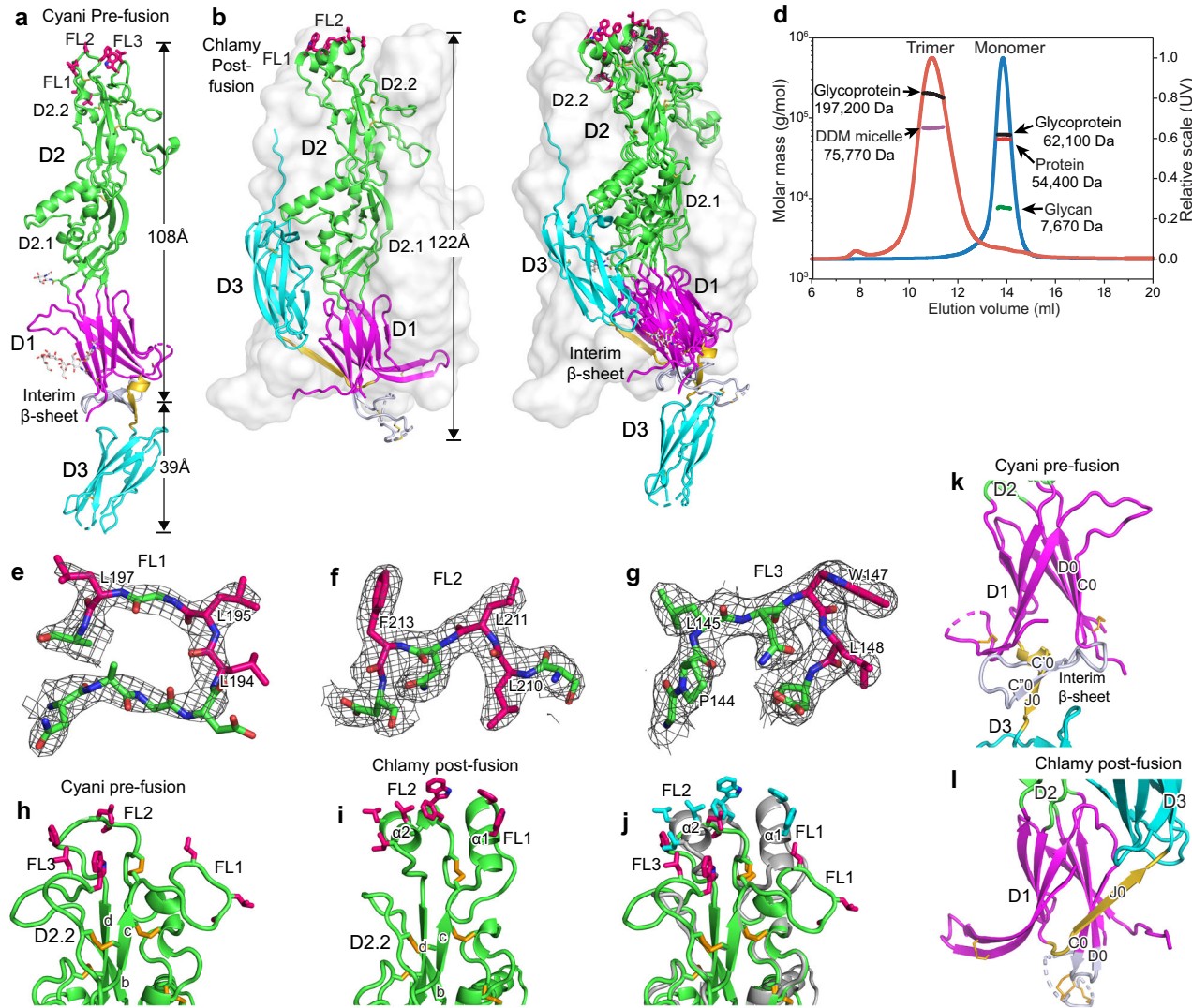

**Fig. 2 Overall Cyani HAP2 structure and comparison to Chlamy HAP2. a** Monomeric Cyani HAP2 shown as ribbon cartoon and colored by domain. Apically exposed residues in the fusion loops, disulfide bonds, and N-linked glycans are shown in sticks with carbons colored hotpink, gold, and white, respectively. The J0 β-strand and the linker between D1 and D3 are colored gold, and the C0D0 loop and C' and C"O β-strands that form the interim β-sheet with the J0 strand are colored bluewhite. **b** The structure of postfusion, trimeric Chlamy HAP2, with chain A colored as in **a** and chains B and C as transparent solvent-accessible surfaces. **c** Comparison of the HAP2 trimeric (Chlamy) and monomeric (Cyani) states after superposition on D1 using DeepAlign[23]. **d** SEC-MALS of monomeric Cyani HAP2 at pH 7.5 (blue curve) and trimeric Cyani HAP2 formed at pH 1.5 with DDM (red curve). Molar masses of the protein, glycan, glycoprotein, and β-DDM components as shown. **e–g** Fusion loops FL1(190–198), FL2 (209–214), and FL3 (143–149) shown in stick with black mesh showing the 2Fo-Fc map contoured at 1σ. **h–j** HAP2 D2.2 of Cyani (**h**) and Chlamy (**i**) alone and superimposed (**j**). Cyani and Chlamy HAP2 have pink and cyan fusion loop residue sticks and light green and silver cartoon backbones, respectively. **k, l** Cyani (**k**) and Chlamy HAP2 (**l**) D1 after structural alignment with same coloring as in **a**.

state of all class II fusion proteins is trimeric and overall similar in structure, with D1 and D2 lining the trimer axis and D3 folded to extend over D1 and D2 (Fig. 1f)[2,3]. Unlike more divergent viral class II fusion proteins, HAP2 proteins in different species are not only structurally homologous, but also in sequence[1,4–7], e.g., as for HAP2 of the red alga Cyani and the green alga Chlamy (Fig. 3). Chlamy HAP2 is well-characterized functionally and a conformational change is required for formation of a trimeric postfusion state in gamete fusion that has been equivalenced with that seen in structural studies[10]. The Chlamy HAP2 ectodomain, like that of Cyani HAP2, is a monomer in solution; however crystals have only revealed its trimeric postfusion state[4–6]. When monomeric Chlamy HAP2 is incubated with liposomes it binds to them and forms trimers[4]. Similarly, when Chlamy HAP2 is incubated with the mild detergent, β-dodecyl maltose (DDM) it

forms trimers and binds to detergent micelles as shown by size in MALS[5]. To test whether Cyani HAP2 could trimerize in the presence of a hydrophobe that mimicked a membrane environment, we incubated it with DDM at the physiologically relevant Cyani growth pH of 1.5. This treatment stimulated formation of Cyani HAP2 trimers that bound to DDM micelles (Fig. 2d). A homogenous peak in gel filtration showed a glycoprotein component with a mass of 197,200 Da and a DDM component with a mass of 75,770 Da. The glycoprotein mass was approximately threefold larger than the monomer glycoprotein mass of 62,100 Da. As Cyani and Chlamy HAP2 are orthologues, each with ectodomains that are metastable monomers in solution that form postfusion state trimers in the presence of detergents, we conclude that the monomeric state seen in the Cyani HAP2 crystal structure is a prefusion state.

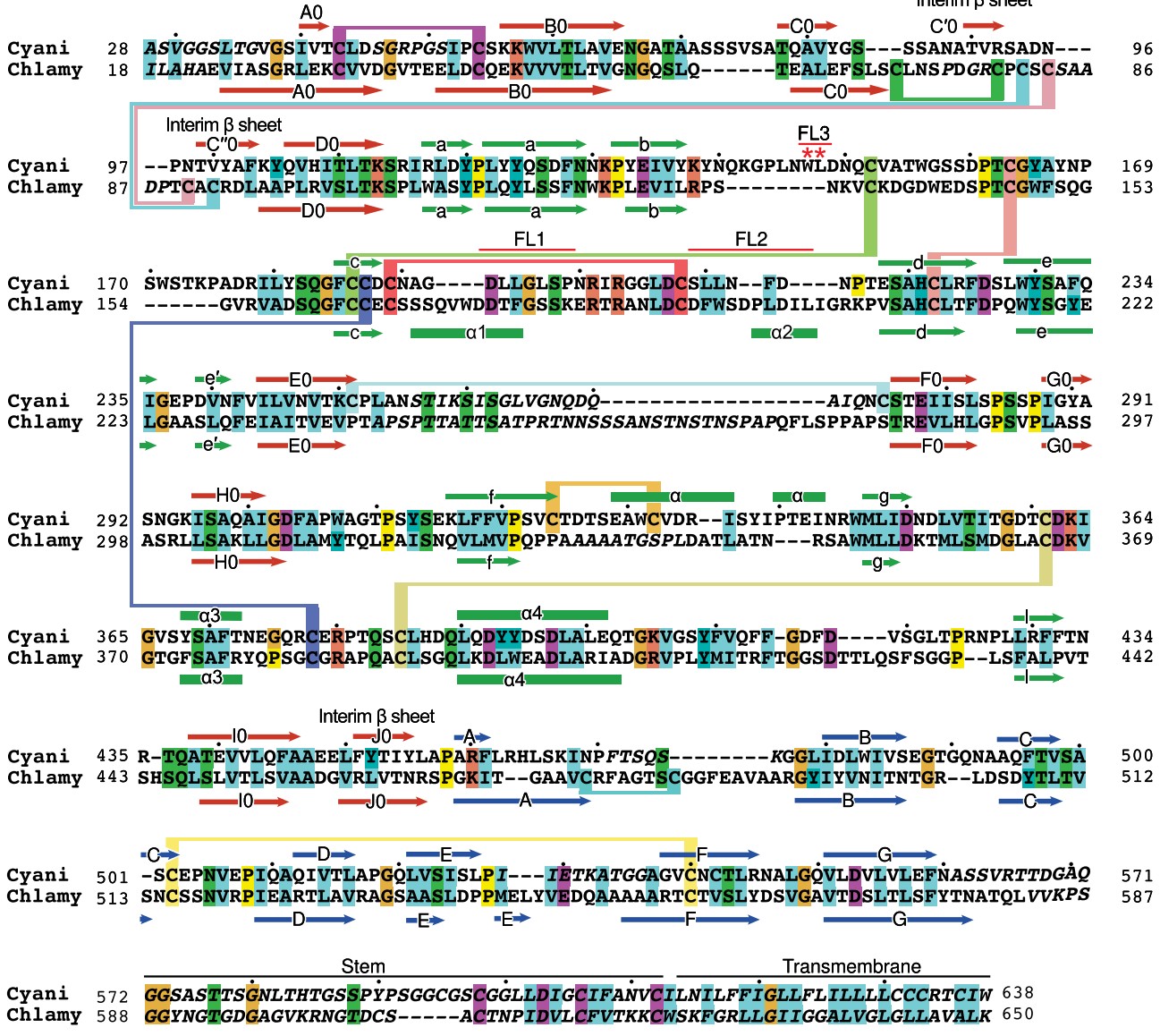

**Fig. 3 Structure-based sequence alignment of Cyani and Chlamy HAP2.** D1, D2, and D3 were individually structurally aligned with DeepAlign[23]. Gaps in poorly aligned regions were closed up using sequence homology. Residues not resolved structurally are in italics. Red asterisks mark putative fusion loop residues.

The fusion loops of Cyani HAP2 locate to the membrane distal end of D2 (Fig. 2a). Electron density for putative fusion loop hydrophobic residues is excellent (Fig. 2e–g). Two fusion loops between the βc and βd-strands in D2.2, FL1 and FL2, are contiguous in sequence but separated by a disulfide-bonded cysteine. Cyani FL1 and FL2 each contain multiple, putative, hydrophobic fusion loop residues (Fig. 2h and Fig. 3). Cyani HAP2 also has putative hydrophobic fusion loop residues in the first portion of the long βb-βc loop, which we designate FL3 (Fig. 2h and Fig. 3). Hydrophobic residues are also found in this region in some viral fusogens and in HAP2 in some other species although not in Chlamy[6]. Most of the FL1 and FL2 fusion loop residues in Chlamy are conserved as hydrophobic residues in Cyani (Fig. 3). However, the conformations of fusion loops differ in Cyani and Chlamy, likely for two reasons (Fig. 2h–j). FL1 and FL2 are four and six residues shorter in Cyani than in Chlamy (Fig. 3), respectively, and thus necessarily differ in loop conformation. Furthermore, apical disposition of the Chlamy hydrophobic fusion loop residues is likely influenced by crystal lattice contacts in which the fusion loops in two trimers contact one another to form hexamers[5].

D2 in HAP2 is extended and is almost identical in length to the combined length of D1 and D3 in the prefusion state (Fig. 2a). Unconventionally in the class II fusogen field, we previously divided D2 into two subdomains, D2.1 and D2.2[5], which are separated by a narrow waist in the middle of D2. There are four connections between D2.1 and D2.2, between the a and b β-strands, the d and e β-strands, and through connections that precede the α3-helix and precede the α4 helix. D2.1 is larger in diameter than D2.2 and D2.2 is thin also in comparison to D1 and D3. D2.2 is stabilized by a single, thin β-sheet with the b, d, and c β-strands and five disulfide bonds that are conserved in Cyani and Chlamy HAP2 (Fig. 3).

D1 has the largest cross-section among HAP2 domains, with two β-sheets, each with four β-strands, that sandwich together over an extensive hydrophobic interface (Fig. 2a). Its large β-structure supports the role of the central D1 domain as a strong fulcrum for conformational change at the D1–D2 and D1-D3 interfaces between the pre and postfusion states (Fig. 2a–c).

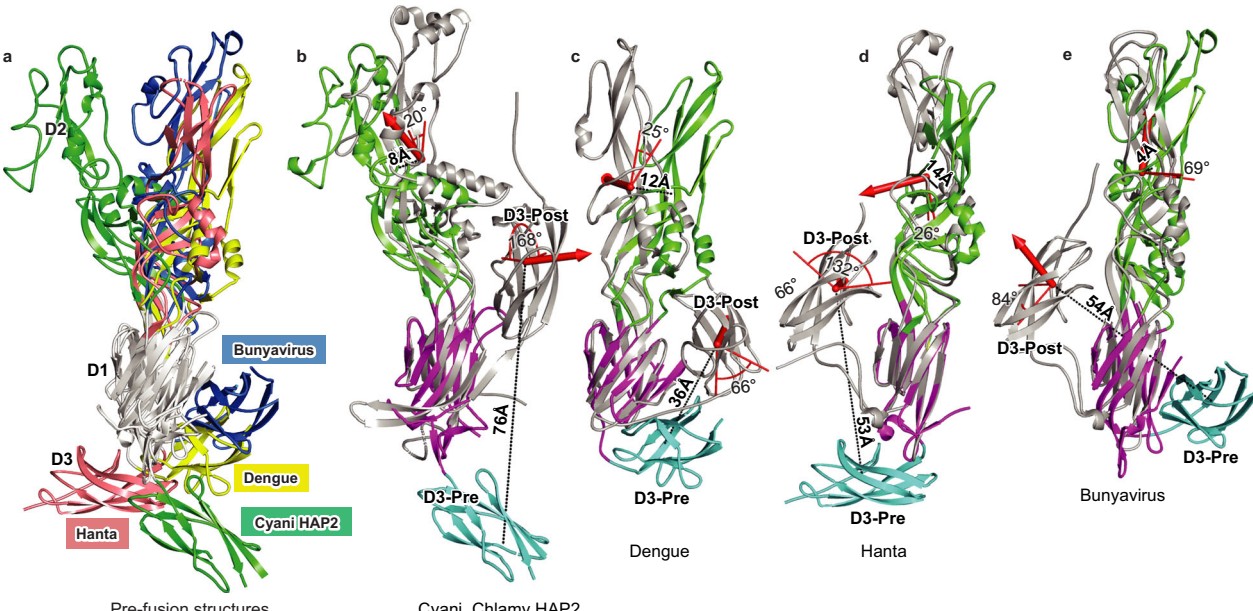

**Fig. 4 D2 and D3 domain reorientation between pre- and postfusion states of HAP2 and viral fusogens. a–e** Structures were superimposed on D1 using DeepAlign[23]. **a** Superimposition of prefusion structures on D1 (colored white) with D2 and D3 colored as shown in key. **b–e** D2 and D3 movements in the postfusion state are shown as distance of center of mass (Å) and as rotation (°) with axis at new center of mass. **b** HAP2 prefusion (PDB 7S0K) and postfusion (PDB 6DBS, chain C). **c** Dengue (PDB 1OKE, and 1OK8). **d** Hantavirus (PDB 5LJZ and 5LJY). **e** Bunyavirus (PDB 4HJ1 and 5J81). Chain A was used unless stated otherwise. The same analysis is shown for D2.1 relative to D1 and D2.1 relative to D2.1 in Supplementary Figure 1A. Using DeepAlign, intact pre and postfusion monomers were aligned with D1 to monomeric Cyani HAP2. Individual D2 and D3 domains were then aligned (1) with prefusion monomers and next (2) with the postfusion monomers to calculate the transformation using the orientation.py script from PyMOL.

Surprisingly, a β-sheet with three β-strands is present in D1 that is absent in the postfusion state of HAP2; we therefore term it the interim β-sheet (Fig. 2a, k). The interim β-sheet is separate from the two β-sheets that form the β-sandwich in D1. Although the interim β-sheet locates to the boundary of D1 with D3, it shares two backbone hydrogen bonds and an extensive hydrophobic interface with other residues in D1 and is therefore part of D1.

The interim β-sheet is formed by D1 segments that are distant in sequence and have different destinies in the HAP2 postfusion state. The first segment, residues 84–105, contains β-strands that follow the C0 strand in D1 that we name C'0 and C''0 (Fig. 2k, l). In the postfusion state of Chlamy HAP2, this segment is incorporated into the D1 β-sandwich and lengthens the C0 and D0 β-strands and the loop between them. The second segment, residues 447–456, becomes in the postfusion state the D1 J0 β-strand, an edge β-strand in the D1 β-sandwich. The J0 β-strand forms the last segment of D1 prior to D3 in the postfusion state. Switching of the J0 β-strand from the interim β-sheet to the D1 β-sandwich in the postfusion state dictates a large change in orientation and position of D3 of HAP2, as the J0 β-strand leads directly into D3.

D3 of the HAP2 prefusion state has a conformation nearly identical to that in previous postfusion and isolated D3 HAP2 structures[4,5,7,11]. It has an Fn3 domain β-sandwich fold with two β-sheets.

Comparison of the pre- and postfusion states of HAP2 and viral class II fusion proteins shows that HAP2 has multiple unique features (Fig. 4a). Among three viral families with both pre and postfusion states, the Flavi, Hanta, and Bunyavirus families, HAP2 has the most extended prefusion state (Fig. 4). Figure 4b–e provides an overview of conformational change of D2 and D3 relative to D1 between pre and postfusion states by showing the distances between the centers of mass of these domains (black dashed lines and distances in Å) and the rotations

about their centers of mass (red cylindrical arrows and arcs at the rotation axis with rotation in degrees). The distance of D3 movement is greatest for HAP2 and its center of mass is positioned more apically (toward D2) than any of the other representative family members compared. The amount of rotation of D3 is also the greatest in HAP2 (168°, coming close to the theoretical maximum of 180°), showing the large effect of the switch of the J0 strand from the interim β-sheet to the β-sandwich. Interestingly, D3 associates with D1 and D2 in the postfusion state in different orientations and in the view of Fig. 4b–e appears to the right in HAP2 and Dengue virus and to the left in Hanta virus and Bunyavirus. Similar comparisons for D2 show markedly different rotation axes relative to D1 (Fig. 4b–e).

**Intermediate and postfusion states of Cyani and Chlamy HAP2.** β-DDM detergent stimulates Cyani HAP2 to form trimers at the physiological growth pH, 1.5, as described above (Fig. 2d). At pH 7.5, β-DDM did not stimulate trimer formation, although the HAP2 peak eluted earlier than in the absence of β-DDM, suggesting that the β-DDM micelle bound to monomeric HAP2 (Fig. 5a). In contrast, trimer formation stimulated by β-DDM was essentially complete at pH 1.5 and 2.0 and somewhat less complete at pH 3.5 (Fig. 5a). At pH 4.3, trimer formation decreased and a peak appeared that was intermediate in position between trimeric HAP2 and monomeric HAP2 in β-DDM at pH 7.5. At pH 5.0, trimer formation further decreased and the intermediate peak became the main peak.

Gel filtration peaks of Cyani HAP2 samples treated with or without β-DDM at different pH values and for differing lengths of time at 37 °C were subjected to negative-stain EM (Fig. 5b–e). Complete class averages are shown in Supplementary Figs. 2 and 3.

The Cyani HAP2 monomer at pH 7.5 in absence of β-DDM is extended and linear in shape, with four distinct densities spanning 155 to 159 Å, and cross-correlates well with the Cyani

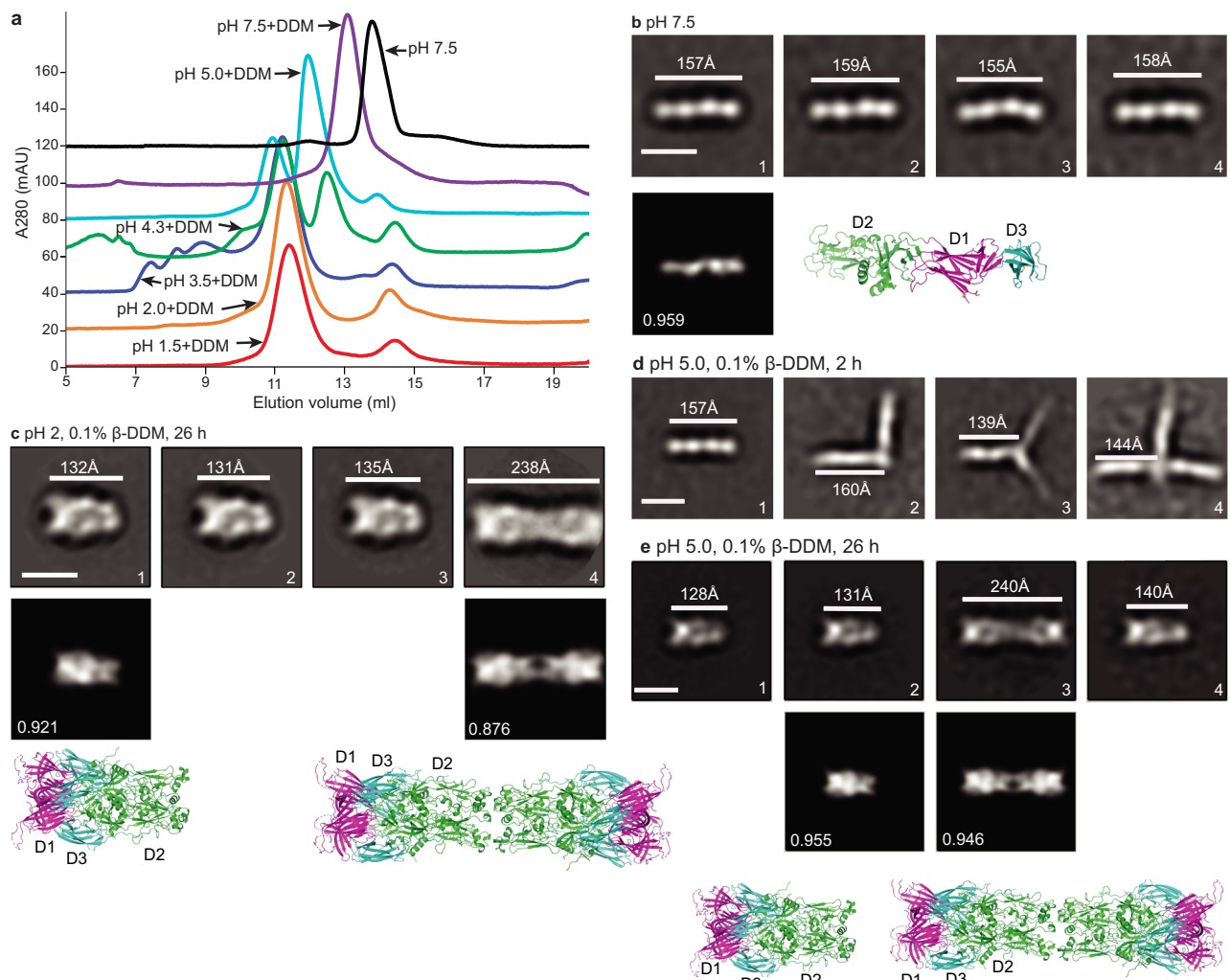

**Fig. 5 Pre, post, and intermediate fusion states of Cyani HAP2. a** Size exclusion chromatography of Cyani HAP2 treated with or without β-DDM at indicated pH values at 37 °C for 26 h. Profiles are separated from one another by successive increases of 20 A280 mAU on the y axis. **b–e** Representative class averages of Cyani HAP2 treated as shown above each panel. Scale bars of 10 nm are shown in left-most class averages. The length of particles (white lines) was measured. Best cross-correlating crystal structure projections and cross-correlation coefficient are shown below selected class averages with corresponding crystal structure orientation as an enlarged ribbon diagram. The experiments of **b**, **d**, **e** were repeated twice and of **c** three times with essentially identical results. All class averages are shown in Supplementary Figures 2 and 3.

HAP2 crystal structure (Fig. 5b). The length in EM agrees with measurements on the crystal structure of 155 Å (Fig. 5b). The four densities in EM agree with our split of D2 into D2.1 and D2.2 subdomains and the equal length in the crystal structure of D2 and D1 + D3.

The Cyani HAP2 trimer formed at pH 2.0 was shorter and wider than monomeric HAP2 (Fig. 5c). The class averages cross-correlated well with 2D projections of the Chlamy HAP2 trimer crystal structure. A population of 5% of the particles resembled the hexamer seen in Chlamy postfusion crystal lattices in which two trimers associate with one another through their fusion loops in D2.2 (Fig. 5c, class average 4). These observations show that the Cyani HAP2 ectodomain can form a postfusion trimer structure in an acidic environment when a hydrophobe is present.

To test whether we could identify intermediates in the formation of the postfusion state, we incubated Cyani HAP2 with β-DDM at pH 5.0 for 2 or 26 h and subjected it to gel filtration and negative-stain EM. At 2 h, several types of intermediate states were present. Branched forms were present with either two or three branches (Fig. 5d). The lengths and the widths of the branches showed that monomers were associating

with one another. At 26 h, trimers predominated, and hexamers were also present (Fig. 5e).

We also examined Chlamy HAP2 for the presence of intermediate states in formation of the postfusion state. We stimulated trimerization with β-DDM at pH 7.5 and examined different time points by gel filtration and negative-stain EM (Fig. 6 and Supplementary Figure 4). Purified Chlamy HAP2 monomer in absence of β-DDM appeared to be an extended monomer with a length of either 110 Å with three densities or 143–150 Å with four densities (Fig. 6a). The shorter class averages would be consistent with flexibility of D3; flexibility can result in loss of density in class averages. One of several minor class averages is shown that is suggestive of two associated monomers; Chlamy HAP2 is prone to trimerize in the absence of detergent. All crystal structures of the trimeric, postfusion state of Chlamy HAP2 were obtained from monomeric protein subjected to crystallization[4,5,7]. Spontaneous hexamer formation by Chlamy HAP2 has also been reported[4]. After 2 h in presence of β-DDM, branched dimers and trimers were present but were rarer than seen with Cyani HAP2 at pH 5 at 2 h, and postfusion-like trimers were also present (Fig. 6b). After 26 h in β-DDM, postfusion

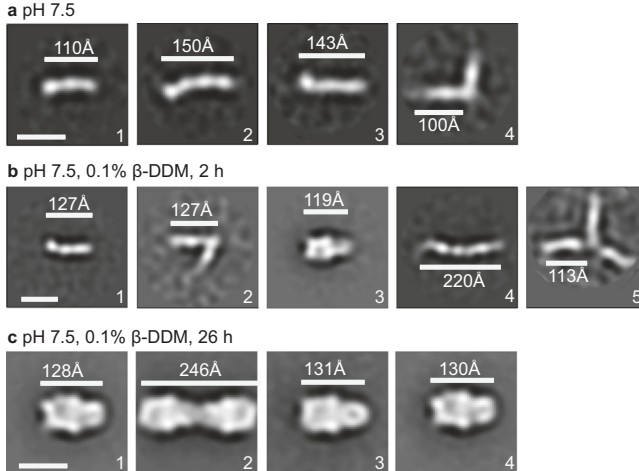

**a** pH 7.5

**b** pH 7.5, 0.1% β-DDM, 2 h

**c** pH 7.5, 0.1% β-DDM, 26 h

**Fig. 6 β-DDM-dependent trimerization of Chlamy HAP2. a** Representative negative-stain class averages of Chlamy HAP2 at pH 7.5 (**a**). **b, c** Representative negative-stain class averages of Chlamy HAP2 at pH 7.5 in the presence of β-DDM after 2 h (**b**) and 26 h (**c**) at 23 °C. The length of particles (white lines) was measured. Scale bars of 10 nm are shown in first class average. Complete class averages are shown in Supplementary Fig. 4. The experiments of **a–c** were repeated twice with similar results.

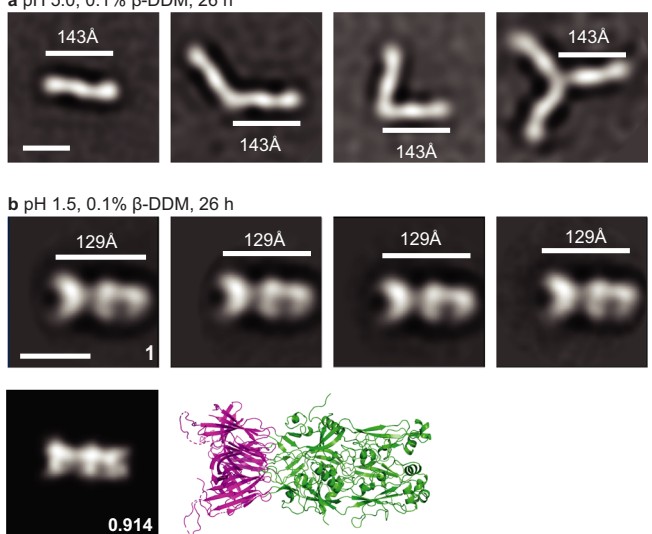

**a** pH 5.0, 0.1% β-DDM, 26 h

**b** pH 1.5, 0.1% β-DDM, 26 h

**Fig. 7 Intermediate and postfusion states of Cyani HAP2 D1D2 fragment. a** Representative class average of Cyani HAP2 D1D2 at pH 5.0 in 0.1% β-DDM after 26 h at 37 °C. **b** Representative class averages of Cyani HAP2 D1D2 at pH 1.5 in 0.1% β-DDM after 26 h at 37 °C. The length of particles (white lines) was measured. Scale bars of 10 nm are shown in the left-most class average. The best cross-correlating crystal structure projection and the cross-correlation coefficient are shown below one class average and the corresponding crystal structure as an enlarged ribbon diagram. Complete class averages are shown in Supplementary Fig. 5. The experiments of **a**, **b** were repeated twice with similar results.

Chlamy HAP2 trimers were the major population (Fig. 6c), along with hexamers (Fig. 6c, panel 2).

To test the prediction that monomers were associating at their D2.2 end rather than their D3 end, and to further study intermediates, we deleted D3 from Cyani HAP2. The resulting Cyani HAP2 D1D2 construct was well expressed. Negative-stain EM showed that some Cyani HAP2 D1D2 remained monomeric after β-DDM treatment at pH 5.0, with three clear densities

corresponding to D2.2, D2.1, and D1 (Fig. 7a, Supplementary Figure 5). The majority of class averages showed two monomer branches and a minority showed three branches. Thus, the first step in formation of trimers is association at the D2.2 end bearing the fusion loops. At pH 1.5 at 26 h, many particles progressed to complete trimer formation, with association along the lengths of D2 and D1 (Fig. 7b). The structure of these trimers was similar to that formed by the complete HAP2 ectodomain, as shown by cross-correlation with the Chlamy crystal structure with D3 removed. Thus, domains D1 and D2 are sufficient for the formation of the core of the HAP2 trimer.

We measured the thermal stability of Cyani HAP2 as a function of pH using intrinsic tryptophan fluorescence (Fig. 8a). Cyani HAP2 was stable over a broad pH range from 1.5 to 7.5, with midpoint temperature of thermal unfolding ($T_m$) values of 54–65 °C, with onset values of denaturation at >43 °C, higher than the temperature of 37 °C at which we measured trimerization stimulated by DDM. Maximal stability was found at pH 4.3 and 5.0.

We next examined the effect of fusion loop mutations on trimer formation by the Cyani HAP2 ectodomain. Mutant proteins were transiently expressed in Expi293F cells, and had yields and appearance in SDS-PAGE similar to WT (Fig. 8b). We first examined trimerization at a pH at which WT HAP2 was highly stable, pH 4.7. All samples showed a single monomeric peak in absence of DDM in running buffer at pH 4.7 (Fig. 8c). When samples were mixed with DDM and subjected to gel filtration in pH 4.7 running buffer containing DDM either immediately, or after 2 h at 37 °C (Fig. 8d, e), WT HAP2 partially formed trimers without incubation and was largely trimeric after 2 h incubation. In contrast, mutations in individual fusion loops, and combined mutations of all three loops, showed no trimerization. We further extended studies to lower pH values of 4.3 and 3.5. No trimer association occurred in absence of DDM. Trimerization of the WT Cyani HAP2 ectodomain was complete at pH 4.3 both after 2 and 4 h in DDM (Fig. 8f–h). The FL2 and FL3 mutants but not the FL1 and FL123 mutants showed partial trimer formation. At pH 3.5, WT HAP2 showed complete trimer formation in DDM and the FL2 and FL3 mutants showed partial trimer formation, whereas the FL1 and FL123 mutants showed no trimer formation (Fig. 8j). These results show that association of fusion loops with a hydrophobic environment is essential for postfusion state trimer formation, that FL1, FL2, and FL3 each contribute to this process, and that FL1 is more important for trimer formation than FL2 and FL3.

To obtain a model of the postfusion state of Cyani HAP2, we used AlphaFold-Multimer[12]. For validation, we first predicted a monomeric structure with AlphaFold ColabFold[13]. Compared to the Cyani crystal structure (Fig. 9a), the model was remarkably accurate, contained the interim β-sheet, and had Cα atom RMSDs of 0.5 to 0.6 Å for individual D1, D2, and D3 domains and an RMSD of 3.4 Å over all 466 residues of the prefusion state (Fig. 9b). Predictions of monomeric structures of Chlamy and *A. thaliana* HAP2 yielded postfusion-like conformations with a hairpin bend between D1 and D3, perhaps influenced by their deposited trimeric structures in the PDB, in contrast to the extended prefusion-like conformation of Chlamy HAP2 seen in EM here and predicted by AlphaFold for Cyani HAP2.

When challenged to predict a trimeric Cyani HAP2 structure, AlphaFold-multimer produced a postfusion-like conformation with a hairpin bend between D1 and D3 (Fig. 9c). The interim β-sheet was absent, and the J0 β-strand was incorporated in D1, as seen in the Chlamy postfusion structure. The conformation of FL3 was almost identical to that in the monomeric crystal structure and the position of FL2 was similar (Fig. 9d). In contrast, the orientation of FL1 became much more apical with

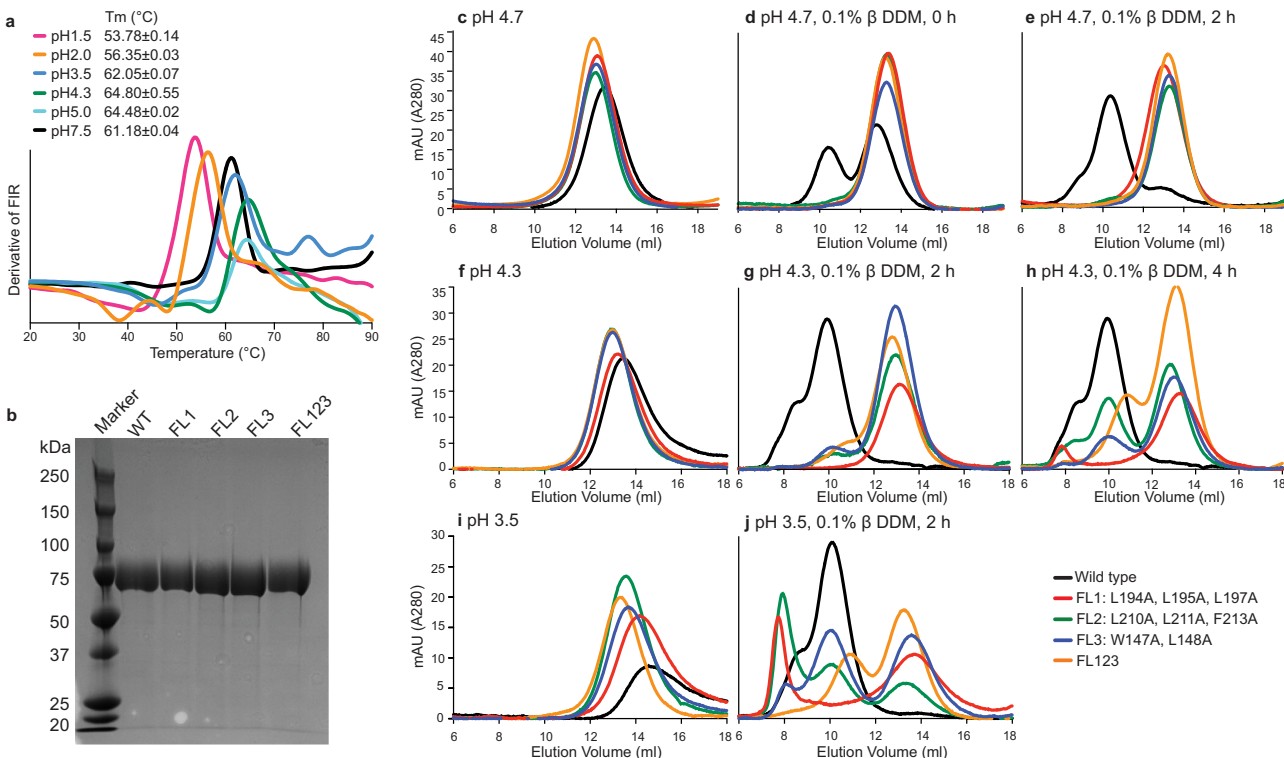

**Fig. 8 Stability of Cyani HAP2 ectodomain to pH and requirement of fusion loop hydrophobic residues for detergent-dependent trimerization.** **a** Temperature-dependent unfolding of HAP2 ectodomain measured by tryptophan fluorescence intensity ratio (FIR) at indicated pH values. The first derivative is plotted. Tm values are mean ± difference from the mean. **b** Purified proteins (3 µg) were subjected to SDS 10% PAGE and Coomassie blue staining. SDS-PAGE of Cyani HAP2 wild type and mutant ectodomains protein was conducted three times with essentially identical results. **c–j** Size exclusion chromatography of Cyani HAP2 wild type and mutant ectodomains treated with or without 0.1% DDM in buffers at the indicated pH values for the indicated times at 37 °C and subjected to size exclusion chromatography in buffers of the indicated pH values in absence or presence of 0.01% DDM, respectively.

putative fusion loop residues L-194, L-195, and L-197 in positions much more appropriate for insertion into a target gamete membrane. Interestingly, the conformation of FL1 became very similar to that seen in the postfusion state of Chlamy HAP2, with the four-residue shortening of FL1 in Cyani relative Chlamy (Fig. 3) accommodated by a shortening of the α1-helix (Fig. 9e).

## Discussion

HAP2 is widely distributed in eukaryotes and is the only currently known eukaryotic gamete fusogen. We have defined the prefusion state of HAP2 from an extremophile red alga that grows optimally at pH 1.5 and 45 °C. Cyani HAP2 shows important differences from viral class II fusion proteins. The structural features revealed here of the HAP2 prefusion state will serve as an important guide for vaccines to HAP2 from other species, including to *Plasmodium* spp. to block malaria transmission. Such vaccines need to recognize the prefusion rather than the postfusion state. Recently, a monoclonal antibody to D3 of *P. berghei* HAP2 was shown to block transmission in mosquitoes[11].

The prefusion state of Chlamy HAP2, like Cyani HAP2, was found to be monomeric in solution[4,5]. Cyani HAP2 also behaved like a monomer in the crystal lattice, with no biologically relevant associations with other monomers. Viral class II fusion proteins are stored in many different types of lattices on the virion surface prior to fusion, including in association with symmetry-related class II fusion proteins or other classes of viral proteins. In these lattices their fusion loops are buried[14–17]. By analogy to viral fusogens, it might be thought that HAP2 should also exist as a dimer or in association with another protein. The monomeric state of Cyani HAP2 in solution and crystals suggests that in the

absence of a hydrophobic environment, such as provided by a detergent here, it does not multimerize. However, we cannot rule out association of the prefusion state of HAP2 with other proteins. Mechanisms that block premature triggering of trimerization may differ for viral class II fusion proteins and HAP2, since HAP2 may be stored intracellularly prior to gamete fusion, or in the case of algae or plants, hidden by the cell wall, digestion of which is required for gamete fusion[10]. Thus far, our best structural knowledge on HAP2 is from red and green algae and flowering plants, all of which have cell walls. Given the diversity of Eukaryota, as well as Archaea in which HAP2 was recently found[18], it is possible that multiple mechanisms may have evolved to prevent HAP2 fusion loops from prematurely triggering fusion[2,3].

The prefusion state of Cyani HAP2 is unusually extended compared to prefusion states of structurally homologous viral class II fusogens. EM showed that the prefusion state of Chlamy HAP2 is also highly extended and rod-like. Furthermore, both Chlamy and Cyani HAP2 exhibited extended prefusion states with four globules in the rod, consistent with the subdivision of domain 2 into D2.1 and D2.2 subdomains. Not only was the prefusion state highly extended, but comparisons of prefusion and postfusion states of HAP2 to those of class II fusion proteins from three viral families showed that the rotational and translational movements of D3 between the pre and postfusion state relative to D1 were also greater in HAP2.

A unique feature of the Cyani HAP2 prefusion state is the presence of an interim β-sheet in D1 that is separate from the two β-sheets in D1 that form the β-sandwich. The interim β-sheet is comprised of two β-strands, C'0 and C"0, that have not previously

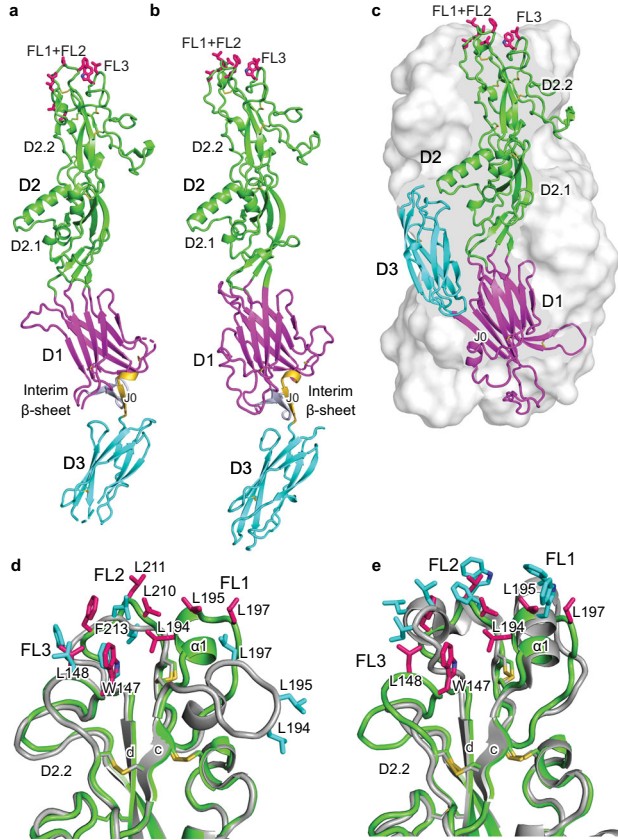

**Fig. 9 Cyani monomeric and trimeric HAP2 structures predicted by AlphaFold. a** Crystal structure of monomeric Cyani HAP2 structure. The cartoon representation is colored by domain with the putative membrane-inserting fusion residues in FL1, 2, and 3 shown in stick and colored hotpink. Disulfide bonds shown as gold stick. Interim β-sheet β-strands are colored gold (J0) and bluewhite (C'0 and C"0 and their loop). **b, c** Monomeric Cyani HAP2 (**b**) and trimeric Cyani HAP2 (**c**) structures predicted by AlphaFold and AlphaFold-Multimer, respectively. Coloring is as in **a**, except in **c** two monomers are shown as solvent-accessible, transparent surfaces. **d, e** Fusion loop comparisons after superposition on D2.2. The AlphaFold trimer model is shown with green backbone and hotpink fusion loop residues in stick superimposed on the Cyani Hap2 crystal structure (**d**) and the Chlamy Hap2 crystal structure (**e**) with silver backbone and cyan fusion loop residues. The Cyani HAP2 monomer was predicted by AlphaFold[13] Colab [https://colab.research.google.com/github/sokrypton/ColabFold/blob/main/beta/AlphaFold2_advanced.ipynb]. The Cyani HAP2 trimer was predicted with AlphaFold-Multimer[12] (https://github.com/deepmind/AlphaFold), run on the cosmic server [https://cosmic2.sdsc.edu:8443/gateway/home.action]. See Supplementary Figure 6 for prediction reliability.

been described in class II viral fusogens, and the J0 β-strand. Among 16 other diverse class II fusion structures including those in Fig. 4, we found D1 J0 β-strands only in two prefusion structures, Chikungunya virus (3N40) and Bunyavirus (4HJ1), and three postfusion structures, EFFI (4OJC), Rift Valley fever virus (6EGT), and tick-borne encephalitis virus (1URZ). In place of J0, the other structures have linkers leading to D3. The J0 β-strand in the Bunyavirus prefusion state is present in an interim β-sheet, but it differs from that in Cyani HAP2 in associating with A0-like and B0-like β-strands rather than β-strands that form between the C0 and D0 strands. The J0 β-strand in Bunyavirus also differs in orientation by ~90°. In the postfusion state, the Bunyavirus interim β-sheet disappears, and the J0 β-strand forms a linker on the opposite side of D1 from the J0 β-strand in the

Chlamy and predicted Cyani HAP2 postfusion states, consistent with D3 hairpin formation on the opposite side of D1 from Cyani HAP2 (Fig. 4b, e). It is interesting that the orientation of D3 relative to the trimer core can differ so much among class II fusion proteins and that in HAP2 the incorporation of the J0 β-strand into the D1 β-sandwich dictates D3 orientation.

We identified intermediate states in the fusion process in which two or three HAP2 monomers associate through their tips at D2.2, to which the fusion loops localize. Identification of these intermediates was facilitated by slowing formation of the postfusion state of Cyani HAP2 at pH 5. At this pH in DDM, branched dimers and trimers associated at monomer tips were abundant at 2 h, whereas at 26 h, postfusion trimers were predominant. Formation of post-fusion trimers in DDM was complete at pH 1.5 and 2.0 and did not occur at pH 7.5 after 26 h. Similar intermediates were seen with Chlamy HAP2 at pH 7.5, but were rarer. To the best of our knowledge, no intermediates in formation of class II postfusion states have previously been trapped. However, it has been proposed that association of D1 and D2 along the trimer axis precedes foldback of D3, because the Dengue fusogen can form postfusion state-like trimers when D3 is deleted[8]. Cyani HAP2 lacking D3 also formed postfusion state-like trimers; formation of branched dimers and trimers by this construct further showed that monomers associate at the tip of D2.2.

Mutations of the three hydrophobic, putative fusion loop residues in FL1 and FL2 and the two such residues in FL3 showed that all three fusion loops contributed to formation of the trimeric postfusion state, which required the presence of DDM to provide a hydrophobic environment. Use of pH values of 4.7, 4.3, and 3.5 showed that trimerization of WT HAP2 was partial at pH 4.7 and complete at pH 4.3 and 3.5. Fusion loop mutants showed no trimerization at pH 4.7. Mutants FL2 and FL3 showed partial trimerization at pH 4.3 and greater but still partial trimerization at pH 3.5, while FL1 and FL123 mutants showed no trimerization. Thus, all three fusion loops contribute to formation of the postfusion state, but use of different pH values showed that FL1 was more important than FL2 and FL3. The importance of association of the fusion loops with a hydrophobic environment for trimerization agreed with negative-stain EM results showing that association of mono-mers through their fusion loops tips into dimers and trimers pre-ceded formation of the postfusion trimer state.

Based on these findings, we propose multiple steps in the conversion of the prefusion to postfusion state of HAP2, in which the key triggering event is membrane association of HAP2 monomers through their fusion loops (Fig. 1). After male and female gamete plasma membranes are brought into close proxi-mity, which requires upstream molecular and physiological processes[10], and the prefusion state of HAP2 is present on the male gamete membrane, the fusion loops of individual monomers insert into the outer leaflet of the female gamete plasma mem-brane (Fig. 1a) and then associate at their fusion loops into dimers and then trimers (Fig. 1b, c). Our EM and fusion loop mutagenesis data favor initial association at the fusion loops rather than along the lengths of D2 and D1, and that association at the fusion loop tips is followed by association over the lengths of D2 and D1, likely by zipping up along the threefold trimer axis from D2.2 to D2.1 to D1 (Fig. 1d). Foldback of D3 follows in which the male and female plasma membranes are pulled closer (Fig. 1e). The large 168° rotation and 76 Å translation of D3 in HAP2 would involve the dissolution of the D1 interim β-sheet and the joining of its J0 β-strand to the edge of a β-sheet in the β-sandwich. As generally postulated in the class II fusion mechanism for viruses and HAP2[2–4], pulling owing to D3 fold-back and destabilization of the female gamete membrane by the fusion loops would bring about fusion of the two gamete mem-branes with formation of a fusion pore to create the zygote

(Fig. 1f). Viral class II postfusion state formation might also proceed through intermediates in which prefusion states initially bind to membranes and associate through the fusion loops at their D2.2 tips.

## Methods

**Expression and purification of HAP2 ectodomain**. Cyani HAP2 amino acid sequence numbering is from RefSeq accession XP_005536505.1. Codon-optimized synthetic cDNA encoding Cyani HAP2 ectodomain (residues 28–564) was cloned into a modified *Drosophila* S2 expression ET15S2 vector with a C-terminal His-tag and transfected into *Drosophila melanogaster* Schneider S2 cells (ExpreS2 cells, ExpreS2ion Biotechnologies), using EXpreS2 transfection reagent. Stable transfectants were selected in EX-CELL 420 Serum-Free Medium (Sigma) supplemented with 4 mg/ml G418 and expanded in EX-CELL 420 medium at 25 °C. Cyani HAP2 D1D2 (residues 28–457) constructs and Cyani HAP2 ectodomain (residues 28–564) fusion loop mutants (FL1: L194A, L195A, L197A; FL2: L210A, L211A, F213A; FL3: W147A, L148A; and combinations of all in FL123) and WT control constructs were cloned into PD2529 expression vector (ATUM) with a C-terminal His-tag for expression in ExpiF293 cells (Thermo Fisher Scientific). 0.8 mg plasmid was incubated with 0.8 mL FectoPRO (Polyplus) in Opti-MEM (Gibco) for 10 min at room temperature and then added to 1 L of Expi293 cells cultured in Expi293 medium (Thermo Fisher Scientific). 24 hrs after transfection, valproic acid and glucose were added to the culture to final concentrations of 3 mM and 0.4% w/v, respectively. Six days after transfection, secreted proteins were purified from the media by His-tag affinity chromatography.

All ectodomain fragments and mutants were purified identically from S2 and Expi293F culture supernatants. After centrifugation at 5,000 g for 20 min, 1 L culture supernatant was filtered (3 micron) and made 1 mM in NiCl₂ and 300 mM in NaCl and applied to a 10 ml Ni-NTA Agarose (Qiagen) affinity chromatography column. After washing with 15 mM, 20 mM and 30 mM imidazole in 20 mM Tris-HCl, pH 7.5, 500 mM NaCl, protein was eluted with 300 mM imidazole in the same buffer. Fragments were then subjected to gel filtration on a Superdex 200 10/300 GL column in 20 mM Tris-HCl, pH 7.5, 500 mM NaCl as described[5]. Proteins were concentrated and frozen at −80 °C in the same buffer. Yields for S2 expression of the Cyani HAP2 ectodomain were 1.5–2 mg/L culture supernatant. Yields for transiently expressed Cyani HAP2 D1D2 was 3–4 mg/L culture supernatant. Yields for transiently expressed WT and FL mutant HAP2 ectodomains were 9–10 mg/L. WT and mutant fusion loop ectodomain fragments (residues 28–564) from Expi293F cells were only used to test the effect of fusion loop mutations on trimer formation (Fig. 8). Other experiments and crystallization with ectodomain fragments were with proteins from S2 cells while experiments with the D1D2 fragment were with proteins from Expi293F cells. No differences in behavior of these proteins were noted, although differences in N-glycosylation between Expi293F cells and S2 cells might have small effects.

**Crystallization and structure determination**. Crystals were grown at 20 °C by hanging-drop vapor diffusion with equal volumes of Cyani HAP2 ectodomain (5–6 mg/ml) and reservoir solution (19% isopropanol, 19% PEG3350, 5% glycerol, 0.1 M pH 6.4 sodium citrate). Larger crystals, obtained after cleavage of N-glycans with Endo D using the same reservoir solution, were used for diffraction data collection. HAP2 ectodomain (1 mg in 1 mL) was cleaved with 500 or 750 units of Endo D (New England Biolabs, 50,000 units/mL) at 4 C° overnight, followed by Superdex 200 gel filtration in 20 mM Tris-HCl, pH 7.5, 500 mM NaCl and concentration. Crystals were cryo-protected with reservoir solution containing 10% and then 15% glycerol and plunged in liquid nitrogen. Data were collected at 100°K on GM/CA beamline 23IDB at the Advanced Photon Source (Argonne National Laboratory) and processed with XDS (version June 1, 2017 BUILT = 20170601)[19]. The structure was solved by molecular replacement with MR-Rosetta in the Phenix suite (version 1.20.1-4487-000)[20] using PDB ID 6DBS[5] as search model. Structures were refined with PHENIX, built with Coot (version 0.9.8.1 EL (ccp4))[21] and validated with MolProbity 4.2 (http://molprobity.biochem.duke.edu/)[22].

Figures were made with PyMOL (version 2.4.1). Structures were superimposed using DeepAlign[23].

**Trimerization of HAP2**. Purified Cyani HAP2 monomer or Cyani HAP2 D1D2 (10–15 mg/ml, 50 μL) in 20 mM Tris-HCl, pH 7.5, 500 mM NaCl was incubated with 450 μL of buffers described below in presence of 0.1% β-DDM, 500 mM NaCl at 37 °C for varying times. The concentration of β-DDM used, 0.1%, was chosen by testing concentrations from 0.03 to 0.6% at pH 1.5 to find the lowest concentration that gave well resolved trimer peaks in gel filtration. Buffers were HCl-KCl (pH 1.5 and 2.0), 50 mM citric acid-trisodium citrate (pH 3.5–5.0) and 20 mM Tris-HCl (pH 7.5). Samples were subjected to size exclusion chromatography using a Superdex 200 10/300 GL column with running buffers of the same pH, 500 mM NaCl, and 0.01% β-DDM and used immediately in negative-stain EM.

To test the effect of fusion loop mutations on trimer formation in gel filtration, WT and mutant Cyani HAP2 ectodomains (16 μM) were incubated with nine volumes of 50 mM citric acid-trisodium citrate buffers at pH 4.7, 4.3, or 3.5, 500 mM NaCl with or without DDM at 37 °C for varying times and gel filtered as described above.

Purified Chlamy HAP2 monomer (10–15 mg/ml, 50 μL) in 20 mM Tris-HCl, pH 7.5, 500 mM NaCl was incubated with 450 μl 0.1% β-DDM in the same buffer at 23 °C for varying times and subjected to Superdex 200 chromatography in 20 mM Tris-HCl, pH 7.5, 500 mM NaCl, 0.01% β-DDM and used immediately in negative-stain EM.

**Negative-stain EM**. Peak fractions were adsorbed to glow-discharged carbon-coated copper grids and stained with freshly prepared 0.75% uranyl formate. Images were acquired on an FEI Tecnai-12 transmission electron microscope at 120 kV. Particles were picked interactively and subjected to 2D alignment, classification and averaging using RELION 3.1[24]. Alignment of class averages with one another, centering, and cross-correlation with crystal structures was done using EMAN2[25]. Selected 2D class averages of Cyani HAP2 trimer were masked and cross-correlated with 2D projections generated at 2° intervals from electron density maps filtered to 25 Å of Chlamy HAP2 trimer crystal structure (PDB ID: 6DBS) or hexamer formed by trimers interacting through fusion loops in the crystal lattice. Particle numbers are shown on bottom of each class average. Classes are ranked from most to least populous from left to right and then from top to bottom. Scale bars show as white lines.

**Multi-angle light scattering**. Cyani HAP2 monomer or trimer was subjected to gel filtration on a Superdex 200 10/300 GL column (GE Healthcare Life Sciences) using an Agilent liquid chromatography system equipped with a DAWN HELEOS II multi-angle light scattering detector, an Optilab T-rEX refractive index detector, and UV detector (Wyatt Technology Corporation). Data were processed in ASTRA 6 using the protein conjugate model. For Cyani HAP2 monomer, $dn/dc$ values of 0.185 and 0.145 ml/g were used for protein and glycan, respectively[26,27], and A280 extinction coefficient of 1.411 ml mg⁻¹ cm⁻¹ was calculated from amino acid sequence. For Cyani HAP2 trimer, $dn/dc$ value of 0.133 was used for the β-DDM component[28]. We calculated the $dn/dc$ value (0.181) of the glycoprotein component of the trimer by using the weight fraction of protein ($f_{prot} = 0.877$) and glycan ($f_{glycan} = 0.123$) of the Cyani HAP2 monomer as $\left(\frac{dn}{dc}\right)_{glycoprot} = \left(\frac{dn}{dc}\right)_{prot} * f_{prot} + \left(\frac{dn}{dc}\right)_{glycan} * f_{glycan}$, described in[5]. Similarly, the extinction coefficient for the of the Cyani HAP2 monomer is 1.237 ml mg⁻¹ cm⁻¹ glycoprotein, which was calculated as $\varepsilon_{glycoprot} = \varepsilon_{prot} * f_{prot} + \varepsilon_{glycan} * f_{glycan}$.

**Stability to pH**. Cyani HAP2 ectodomain purified from Expi293F cells (20 mg/mL in 1 μL 20 mM Tris-HCl, pH 7.5, 500 mM NaCl was diluted with 19 μL of buffers containing 500 mM NaCl and the following: 50 mM HCl-KCl (pH 1.5 and pH 2.0), citric acid-trisodium citrate (pH 3.5, pH 4.3, and pH 5.0), or Tris-HCl (pH 7.5). Duplicate samples in nanoDSF Grade Standard Capillaries (NanoTemper Technologies) were subjected to thermal unfolding in a Prometheus NT.Plex nanoDSF instrument (NanoTemper Technologies) with a linear thermal ramp of 2 °C/min from 20 °C to 95 °C with excitation at 275 nm at a power of 30%. The fluorescence intensity ratio (FIR) of tryptophan emission at 350 nm/330 nm was determined.

**Reporting summary**. Further information on research design is available in the Nature Research Reporting Summary linked to this article.

## Data availability

The atomic coordinates and the structure factors for Cyani HAP2 have been deposited in the PDB under the accession number 7S0K. Monomeric and trimeric Cyani HAP2 structures predicted with AlphaFold have been uploaded to Zenodo, accession code 6476359 with https://doi.org/10.5281/zenodo.6476359. Source data are provided in this paper.

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

## Acknowledgements
This work was supported by NIH grant R01-HL148755 (T.A.S.). We thank the staff of beamlines GM/CA 23IDB at the Advanced Photon Source (Argonne National Laboratory) for support. We thank Margaret Nielsen for the illustrations. We thank Kelly L. Arnett at the Center for Macromolecular Interactions of Harvard Medical school for training and consultation on SEC-MALS measurement. We thank Dr. Debora Marks and Joseph Min for helping with an earlier AlphaFold prediction of the trimeric Cyani HAP2 structure.

## Author contributions
J. F. contributed to the conception and experimental design, carried out the biochemical studies and crystallization, solved, and refined the structure, wrote the draft, and edited the manuscript; X. D. contributed to structure refinement; Y. S. contributed to EM cross-correlation; C. L. supervised cloning and protein expression, contributed to discussions and edited the manuscript. T. A. S. conceived and supervised the project, contributed to methodology, experimental design, and structure refinement, and wrote the manuscript.

## Competing interests
The authors declare no competing interests.
