## [Peer Review File · Nature Communications]

Monomeric prefusion structure of an extremophile gamete fusogen and stepwise formation of the postfusion trimeric stateReviewers' Comments:

Reviewer #1:

Remarks to the Author:

Feng et al., report structural analysis of a eukaryotic gamete fusogen, HAP2 from an extremophile alga. A variety of approaches including crystallography, negative stain EM, alphafold structure prediction are applied to characterize various conformational states that HAP2 can adopt. The most novel among the set described in this study, which adds to existing knowledge in the field of type II fusogens, is an extended, monomeric form of HAP2 ectodomain observed by X-ray crystallography and gel filtration chromatography. This conformation is proposed to represent the prefusion form of HAP2 fusogens, and while it indeed is novel, other type II proteins such as EFF1 have been reported to be monomeric in their prefusion forms as well (Zeev-Ben-Mordehai, Nat Comm 2014; PMID 24867324), albeit based upon moderate resolution cryo-electron tomography, in the EFF1 case the fusogen was displayed on a membrane rather than truncated in soluble form as described in the present study. The monomer configurations differ to some extent, especially in the positioning of Domain III. The crystal structure of the HAP2 monomer, however, adds significant new detail in revealing the organization of monomeric HAP2. In addition, the authors report that elevating the solution pH to 5.0, well above the alga's optimal growth conditions of pH 1.5 and 45°C, in the presence of membrane or detergent, leads to formation of "rosettes" or discrete clusters of extended HAP2, which are presumably tethered together by interactions of the hydrophobic fusion loops on Domain II. These are proposed to represent intermediate states of HAP2 that the authors argue would be populated over the course of HAP2-mediated membrane fusion.

The structural biological analysis is sound and the finding of an extended monomeric HAP2 conformation in the crystal structure is interesting indeed. My primary reservations are that while this state is assigned to a prefusion state and the fusion-loop tethered clustered form is assigned as an intermediate state, there is no accompanying biological experiments to show that these structures or organizations indeed correspond to the assigned functional states. They reflect conformational states that the proteins can adopt in the forms analyzed, i.e. as soluble ectodomains, but to call them prefusion and intermediate states is premature and could well be contradicted by structural analysis of the proteins in situ where HAP2 may be arrayed, interacting with other cellular proteins and membrane. My feeling is it must be made clear that these are not definitively assignable to prefusion and fusion intermediate states, but rather are proposed to reflect those states.

The authors also may be interested to consider the data in Cao and Zhang, PNAS 2013 (PMID 23898184) where an extended form of an alphavirus' type II fusion protein is observed connecting virus and liposome surfaces.

Specific items:

The abstract reads for a rather specialized readership and perhaps could be written to better place the work into a broader context that will be appreciated by a general audience.

Reviewer #2:

Remarks to the Author:

The manuscript entitled "Monomeric prefusion structure of an extremophile gamete fusogen and stepwise formation of the postfusion trimeric state" is an integrative structural biology approach towards the understanding of the fusion of membranes by describing different conformational and oligomeric states of the HAP2 protein from an extremophile (Cyani) when compared to the homologue found in green algae (Chlamy). The manuscript attempts to provide a general mechanism of the role of HAP2 in this crucial biological process (membrane fusion) that I consider to be relevant to a broad audience like the readers of Nature Communications. However, the statements in the current manuscript are far too ambitious. The paper severely lacks analysis from the structural side, and the

functional aspects are not considered at all. That being said, several things could be addressed to get it accepted for publication. Therefore, I suggest a major revision addressing the following:

Major points:

- From reading the abstract, one cannot understand what the main findings are nor the purpose of the study. It should be re-written.
- The hypothesis presented in the introduction (page 2, line 38) that the findings from the current study are relevant to Plasmodium sexual reproduction is very vague and highly speculative. Specially the statement about “development of vaccines for malaria”, all these should be removed or presented in another way. For example, the homology of the HAP2 proteins needs to be shown and the AlphaFold models could be shown to speculate on structural similarities.
- I find the mechanism describing the role of HAP2 in fusion (page 13, line 312) to be not robust enough from a structural point of view. For example, the roles of fusion loops in D2 and D1 are not properly described. This being a purely structural study is lacking of structural in-depth comparisons:
 - What parts are seen as flexible in the structures but contributing to the oligomerization based on functional studies present in the literature.
 - Please show the electron density map of the xtal structures for the fusion loops (FL1, 2 and 3).
 - The role of key residues in oligomerization needs to be addressed and residues identified. For example, there should be a mutagenesis study to identify such residues on the fusion loops.
 - What triggers the trimer formation? At least in the case of Cyani it could be proposed. Why is it so dependant on pH? What is the pKa of the residues triggering the conformational changes? Servers like PROPKA or H++ could help.
 - Is the protein stable and properly folded at pH 3.5 and at 7.5? A simple DSF run could be done at different pHs to show that the protein stability for the proposed intermediates is not compromised.
- There is no functional data showing membrane fusion or remodelling. Experiments using membrane models such as GUVs or liposomes could be performed.

Other points:

- AF2 internal quality metric (pLDDT) should be shown in Figure 7 or in a supplementary figure. Additionally, the model must be uploaded to a public repository like Zenodo so readers can analyze it.
- On page 3, line 78: At “intermediate pH values”; please indicate which values.
- A more complete description of methods would be highly valuable, especially in the protein expression and purification section.

Comments to figures:

- Figure 1A would profit from showing overlays between the monomeric and trimeric forms. For Figure 1D: Please show the overlaid figure for Cyani and Chlamy. The positioning of side-chains on the FL1 to 3 needs to be backed up by the electron density maps.
- Figure 3: What domain or section of the protein was used to make Figure 3 structural alignments?
- Figure 4 A. The Y axis is cut .
- Figure 7. Show overlaid structures. AF2 legend needs to be better explained. In addition, the authors should better explain in the methods section which version of AF2 was used (AF2 from DeepMind, ColabFold, LocalColabFold?).
- Figure 8. Not enough data supporting panels E to F. This has to be clarified.

REVIEWER COMMENTS

Reviewer #1 (Remarks to the Author):

Feng et al., report structural analysis of a eukaryotic gamete fusogen, HAP2 from an extremophile alga. A variety of approaches including crystallography, negative stain EM, alphafold structure prediction are applied to characterize various conformational states that HAP2 can adopt. The most novel among the set described in this study, which adds to existing knowledge in the field of type II fusogens, is an extended, monomeric form of HAP2 ectodomain observed by X-ray crystallography and gel filtration chromatography. This conformation is proposed to represent the prefusion form of HAP2 fusogens, and while it indeed is novel, other type II proteins such as EFF1 have been reported to be monomeric in their prefusion forms as well (Zeev-Ben-Mordehai, Nat Comm 2014; PMID 24867324), albeit based upon moderate resolution cryo-electron tomography, in the EFF1 case the fusogen was displayed on a membrane rather than truncated in soluble form as described in the present study. The monomer configurations differ to some extent, especially in the positioning of Domain III. The crystal structure of the HAP2 monomer, however, adds significant new detail in revealing the organization of monomeric HAP2. In addition, the authors report that elevating the solution pH to 5.0, well above the alga's optimal growth conditions of pH 1.5 and 45°C, in the presence of membrane or detergent, leads to formation of "rosettes" or discrete clusters of extended HAP2, which are presumably tethered together by interactions of the hydrophobic fusion loops on Domain II. These are proposed to represent intermediate states of HAP2 that the authors argue would be populated over the course of HAP2-mediated membrane fusion.

The structural biological analysis is sound and the finding of an extended monomeric HAP2 conformation in the crystal structure is interesting indeed. My primary reservations are that while this state is assigned to a prefusion state and the fusion-loop tethered clustered form is assigned as an intermediate state, there is no accompanying biological experiments to show that these structures or organizations indeed correspond to the assigned functional states. They reflect conformational states that the proteins can adopt in the forms analyzed, i.e. as soluble ectodomains, but to call them prefusion and intermediate states is premature and could well be contradicted by structural analysis of the proteins in situ where HAP2 may be arrayed, interacting with other cellular proteins and membrane. My feeling is it must be made clear that these are not definitively assignable to prefusion and fusion intermediate states, but rather are proposed to reflect those states.

The authors also may be interested to consider the data in Cao and Zhang, PNAS 2013 (PMID 23898184) where an extended form of an alphavirus' type II fusion protein is observed connecting virus and liposome surfaces.

Many other prefusion structures of class II fusion proteins have been published. While EFF1 is one such, and is from an eukaryote, no sequence homology is detectable between it and HAP2 fusogens, and thus there is no more similarity of HAP2 with EFF1 than with viral fusogens. The EFF1 tomograms did not show any structural detail aside from the outline of the volume. One monomer from an EFF1 postfusion state was flexibly fit into the density. However, the result is rather odd, because the intact fusogen was not modeled in the orientation that would allow its C-terminal transmembrane domain to embed in shed membrane vesicles. Instead, the orientation chosen places its fusion loops associated with the membrane and its C-terminal transmembrane domain far from the vesicle membranes.

It is correct that we are far from having views of a biological process involving viral -host fusion or gamete fusion, and when we do so, they will likely be lower resolution.

It is well accepted that it is useful to have models of biological processes using purified components. That has been an important basis for biochemical progress ever since the 1800s when enzymes began to be analyzed in cell free extracts.

We call the monomeric state of HAP2 a prefusion state for two reasons. 1. It is clearly distinct from the postfusion state. 2. It can be made in the presence of a hydrophobe to transition to the post-fusion state. We now make these arguments in an early paragraph of Results. We also show new data that at pH 4.7, 4.3, and 3.5, addition of detergent is sufficient to trigger trimer formation, and that mutations in any of the three fusion loops slows or prevents trimerization. These data together with EM make a reasonable case that association of monomers through fusion loops is on pathway towards trimer formation.

We now place the overall model for cell fusion in Figure 1, as a guide to readers to follow the text on the structural transitions described and in part to emphasize that the last part of this model is widely used in the field.

Specific items:

The abstract reads for a rather specialized readership and perhaps could be written to better place the work into a broader context that will be appreciated by a general audience.

Agreed. The Abstract has been rewritten.

Reviewer #2 (Remarks to the Author):

The manuscript entitled "Monomeric prefusion structure of an extremophile gamete fusogen and stepwise formation of the postfusion trimeric state" is an integrative structural biology approach towards the understanding of the fusion of membranes by describing different conformational and oligomeric states of the HAP2 protein from an extremophile (Cyani) when compared to the homologue found in green algae (Chlamy). The manuscript attempts to provide a general mechanism of the role of HAP2 in this crucial biological process (membrane fusion) that I consider to be relevant to a broad audience like the readers of Nature Communications. However, the statements in the current manuscript are far too ambitious. The paper severely lacks analysis from the structural side, and the functional aspects are not considered at all. That being said, several things could be addressed to get it accepted for publication. Therefore, I suggest a major revision addressing the following:

Major points:

- From reading the abstract, one cannot understand what the main findings are nor the purpose of the study. It should be re-written.

It has been.

- The hypothesis presented in the introduction (page 2, line 38) that the findings from the current study are relevant to Plasmodium sexual reproduction is very vague and highly speculative. Specially the statement about "development of vaccines for malaria", all these should be removed or presented in another way. For example, the homology of the HAP2 proteins needs to be shown and the AlphaFold models could be shown to speculate on structural similarities. It is removed.

- I find the mechanism describing the role of HAP2 in fusion (page 13, line 312) to be not robust enough from a structural point of view. For example, the roles of fusion loops in D2 and D1 are not properly described. This being a purely structural study is lacking of structural in-depth comparisons:

Thanks to the Reviewer's suggestions, we have now added mutation of fusion loop residues that shows that they are indeed required for the detergent-stimulated trimerization process.

-What parts are seen as flexible in the structures but contributing to the oligomerization based on functional studies present in the literature.

That is indeed a good question but although it appears fusion loops may reposition to bind the membrane, the density appears good. This is often the case for viral fusion proteins as well.

-Please show the electron density map of the xtal structures for the fusion loops (FL1, 2 and 3).

Now done.

-The role of key residues in oligomerization needs to be addressed and residues identified. For example, there should be a mutagenesis study to identify such residues on the fusion loops.

Yes, excellent suggestion, now done.

-What triggers the trimer formation? At least in the case of Cyani it could be proposed. Why is it so dependant on pH? What is the pKa of the residues triggering the conformational changes? Servers like PROPKA or H++ could help.

The Cyani HAP2 needs to be at acidic pH, to which it is evolutionarily adapted. pH alone is not sufficient and detergent is the trigger, as now shown at pH 4.7, 4.3, and 3.5.

-Is the protein stable and properly folded at pH 3.5 and at 7.5? A simple DSF run could be done at different pHs to show that the protein stability for the proposed intermediates is not compromised.

Good question. We should have done this earlier. This was a helpful suggestion and we have incorporated it.

- There is no functional data showing membrane fusion or remodelling. Experiments using membrane models such as GUVs or liposomes could be performed.

Formation of trimers and intermediates are done with the mild detergent DDM. It is an excellent mimic of the effects of membranes, as shown with Chlamy HAP2; our group previously showed trimer formation in DDM while Rey and Krey showed trimer formation with liposomes. Detergent has the added advantage of allowing facile negative stain EM and image averaging. Looking at membrane fusion would be beyond the scope of this MS. We have clarified above that the model for membrane fusion is from the literature, not from our work.

Other points:

- AF2 internal quality metric (pLDDT) should be shown in Figure 7 or in a supplementary figure. Additionally, the model must be uploaded to a public repository like Zenodo so readers can analyze it.

Now done.

- On page 3, line 78: At “intermediate pH values”; please indicate which values.

Done

- A more complete description of methods would be highly valuable, especially in the protein expression and purification section.

Done.

Comments to figures:

- Figure 1A would profit from showing overlays between the monomeric and trimeric forms. For Figure 1D: Please show the overlaid figure for Cyani and Chlamy. The positioning of side-chains on the FL1 to 3 needs to be backed up by the electron density maps.

Both are done.

- Figure 3: What domain or section of the protein was used to make Figure 3 structural alignments?

D1. Now described in Methods and Figure legend- and the method of DeepAlign is important for structural homologues without sequence homology.

- Figure 4 A. The Y axis is cut .

It is not absolute as each trace is shifted in y axis. However, we have redone the axis.

- Figure 7. Show overlaid structures. AF2 legend needs to be better explained. In addition, the authors should better explain in the methods section which version of AF2 was used (AF2 from DeepMind, ColabFold, LocalColabFold?).

Now done.

- Figure 8. Not enough data supporting panels E to F. This has to be clarified.

This is from the field in general, not from our study, as cited. This has in part been clarified by moving this figure up to Fig. 1 as explained in response to Reviewer 1.